# Effects of LED Light Colors on the Growth Performance, Intestinal Morphology, Cecal Short-Chain Fatty Acid Concentrations and Microbiota in Broilers

**DOI:** 10.3390/ani13233731

**Published:** 2023-12-01

**Authors:** Yihui Liu, Youkuan He, Siqin Fan, Xinyu Gong, Yuqiao Zhou, Yaowei Jian, Jiuyi Ouyang, Qianming Jiang, Peihua Zhang

**Affiliations:** 1College of Animal Science and Technology, Hunan Agricultural University, Changsha 410128, China; yihuiliu2023@hotmail.com (Y.L.); 15736258627@163.com (Y.H.);; 2Department of Animal Sciences, University of Illinois, Urbana, IL 61801, USA

**Keywords:** broilers, feed efficiency, intestinal histology, short-chain fatty acid, cecal microbiome

## Abstract

**Simple Summary:**

The aim of this experiment was to investigate the effects of different LED light colors on the growth performance, intestinal morphology, cecal short-chain fatty acid concentration, and microbiota in broilers. The results showed that blue and blue–green composite light would have better feeding advantages, as demonstrated by the fact that blue and blue–green composite light could improve ileal morphology and affect SCFAs concentration by changing the abundance of genus-level flora to improve growth performance.

**Abstract:**

This study aimed to explore the effects of light-emitting diode (LED) light colors on growth, intestinal morphology, and cecal microbiota in broilers. A total of 360 healthy male Arbor Acres (AA) broilers with similar weights were selected and divided into four groups with six replicates in each group and 15 broilers in each replicate: LED white light (W), LED green light (G), LED blue light (B), and LED blue–green composite light (BG). The experimental period was 42 d, the light cycle of each treatment group was 23L:1D (23 h of light, one hour of darkness) from 1 to 3 d, and the light cycle from 4 to 42 d was 16L:8D; light intensity was 20 Lux. The results showed that the average daily feed intake and final weight of broilers receiving the B group were the highest in 21 d and 42 d compared with other groups. The average daily feed intake of the BG group was lower than that of the B group. In the same light color, small intestine villus height grows with age. On days 21 and 42, compared with other groups, the ileal villus height was higher, the crypt depth was lower, and the V/C ratio (villus to crypt ratio) was higher in the BG group. The combination of blue–green composite light was beneficial to increase the content of propionate, isobutyrate, butyrate, isovalerate, and valerate in the cecum of 21-day-old broilers and the content of isobutyrate in the cecum of 42-day-old broilers, and a decrease in cecal short-chain fatty acid concentrations with age. The B group and the BG group had higher abundances of *Bacteroidetes* at day 21 of age and lower abundances of Phascolarctobacterium at day 42. However, no cecal microbiota differences were detected by the Bonferroni-corrected test. In general, our research results showed that light color could promote the growth of broilers by affecting intestinal morphology, microbiota abundance (needs to be validated by further experiments), and cecal short-chain fatty acid concentrations. And blue and blue–green composite lights are more suitable for broiler growth.

## 1. Introduction

The biological clock adjusts the biological state according to the environment [1]. The exogenous rhythm is regulated by various factors, such as climate, temperature, light, and feed. However, the endogenous rhythm produces rhythmic fluctuations of biological processes that are not affected by exogenous rhythm factors in 24 h [2,3,4]. For the circadian rhythm, the endogenous rhythm and the exogenous rhythm are often synchronized [5].

Light is the most sensitive exogenous factor regulating circadian rhythm and significantly affects physiological and behavioral activities [2]. Broilers are susceptible to light because of the photoreceptor cells in their retinas and the photoreceptors of the deep brain [6]. Light stimulates photoreceptors and transmits information to the suprachiasmatic nucleus (SCN) in the hypothalamus, which interprets retinal light signal information and provides direct and indirect rhythmic cues to the rest of the body [7]. The opsins in the eyes of broilers are sensitive to light color with different wavelengths. Light color stimulation can significantly affect broilers’ growth [8]. Different light colors will have different effects on broilers. The current research showed that blue light and green light are more conducive to the growth of the broilers because green or blue light can significantly promote the myogenic process of satellite cells [9,10]. In addition to monochromatic light, the study of combined monochromatic light showed that white LED supplemental blue/green LED or switching light from green (blue) to blue (green) leads to further weight gain in broilers [11,12].

The function of the gut and the stability of the microflora reflect the health of the digestive tract [13]. Maintaining gut health is critical for promoting body growth. In recent years, the research around intestinal tissue has been a hotspot. The complex neuro-endocrine network called the brain–gut axis connects the brain and the gastrointestinal tract. The brain–gut axis is a bidirectional regulatory axis between gastrointestinal function and the central nervous system [14]. A number of animal and human studies show that intestinal microflora is a key regulator of the brain–gut axis, and short-chain fatty acid (SCFAs) is a potential mediator [15]. Recent studies showed that light exposure changes affected gut microbes’ circadian rhythm [16,17].

According to the above research status, it can be found that most of the current research on the effect of light color in broilers is focused on monochromatic light, monochromatic light converted to another monochromatic light or two monochromatic lights superimposed; there is a lack of research on the composite light. And there is a gap in the research on light regulation of the broiler gut (microbiota and their metabolites).

We wanted to preliminarily investigate whether composite light affects gut microbiota and promotes growth in broilers. The purpose of this study was to provide information on a proper lighting system for the efficient production of broilers. We study how the LED white light, LED green light, LED blue light, and LED blue–green composite light affect the growth performance, intestinal morphology, cecal SCFAs concentrations, and microbiota of broilers, comparing the feeding advantages of different LED lights.

## 2. Materials and Methods

### 2.1. Animals and Experimental Design

The study was carried out in compliance with the National Institutes of Health (Changsha, China) guidelines for the care and use of experimental animals, with approval from the Institution Animal Care and Use Committee of the College of Animal Science and Technology, Hunan Agricultural University (Changsha, China).

In a randomized complete block design, a total of 360 1-day-old healthy male Arbor Acres (AA) broilers were equally distributed by weight of origin into four treatment groups. Broilers were divided into 4 treatments: LED white light (W), LED green light (G), LED blue light (B), and LED blue–green composite light (BG). Treatments were arranged in four isolated spaces, which were demarcated by shading cloth. Each treatment had 90 broilers kept in six replicated pens (15 broilers per pen) with a density of 15 broilers/m^2^ (Stocking densities of 32–35 kg/m^2^ are expected by day 42). The experimental period was 42 days, the light cycle of each treatment group was 23L:1D (23 h of light, one hour of darkness) from 1 to 3 d, and the light cycle from 4 to 42 d was 16L:8D, 20Lux at the level of broilers’ head is. LED (The light-emitting diode) was the only light source and the spectral characteristics involved in this study are shown in Figure 1. The composition of diets is shown in Appendix A.

### 2.2. Growth Performance

Each broiler was weighed and recorded on an empty stomach at 07:00 on the 1st, 21st, and 42nd day of the experiment. Broilers fasted for 12 h while drinking water ad libitum on the day before sampling.

During the experiment, the weight of feed and residual in each pen were accurately weighed weekly. The final body weight (FBW), average daily gain (ADG), average daily feed intake (ADFI), and feed conversion rate (FCR) per broiler for the ages of 21 d and 42 d were calculated.

### 2.3. Intestinal Histomorphology

On the 21st and 42nd day of the experiment, one broiler from each replication close to the average body weight (the average body weight of the groups is shown in Appendix A) of the group was selected and killed by cervical dislocation followed by decapitation (24 individuals per age). After dissection, the duodenum, jejunum, and ileum of the intestinal tissues were collected, fixed with 4% polyoxymethylene for 24 h, embedded in petrolin, and cut into 2–3 µm sections. Following prior methods, the sections were stained with hematoxylin first, then counterstained with eosin, and lastly examined under an optical microscope. On 10 intact, well-oriented villi per intestinal section, the villus height (the distance from the apex of the villus to the junction of the villus and crypt) and crypt depth (the distance from the junction to the basement membrane of the epithelial cells at the bottom of the crypt) were measured, and the villus height/crypt depth (V/C) ratio was calculated.

### 2.4. Cecal Sampling and Short-Chain Fatty Acid Measurement

The killing method was in accordance with the aforementioned details. After dissection, the contents of the cecum were collected, placed in a 5 mL sterile cryopreservation tube, quickly put into liquid nitrogen, and then stored in the freezer at −80 °C.

According to the method described by [18], the concentration of short-chain fatty acids (SCFAs) was measured in each replication, including acetic acid, propionic acid, butyric acid, pentanoic acid, isobutyric acid, and isovaleric acid.

### 2.5. Cecal Microbiota

The killing method and cecal sampling were in accordance with the aforementioned details.

The 338F_—_806R region of the bacterial 16S rRNA gene were conducted to PCR (338F: 5′-ACTCCTACGGGAGGCAGCAG-3′; 806R: 5′-GGACTACHVGGGTWTCTAAT-3′). The PCR components contained 5×FastPfu Buffer (4 μL), 2.5 mM dNTPs (2 μL), 0.8 μL of Forward Primer (5 μM) and Reverse Primer (5 μM), FastPfu Polymerase (0.4 μL), BSA (0.2 μL), Template DNA (10 ng) and supplement ddH_2_O to 20 μL. Cycling parameters were 95 °C for 3 min, followed by 27 cycles at 95 °C for 30 s, 55 °C for 30 s, and 72 °C for 45 s, and a final extension at 72 °C for 10 min. Duplicate PCR products were mixed and further verified with 2% agarose gel electrophoresis. PCR products were recovered using an AxyPrepDNA gel recovery kit (AXYGEN Co., Ltd., San Francisco, CA, USA). PCR products were quantified by reference to preliminary electrophoretic quantification using the QuantiFluor™-ST Blue Fluorescence Quantification System (Promega Co., Ltd., Madison, WI, USA). The library built by NEXTFLEXRapidDNA-SeqKit was carried out for 16s rRNA sequencing by Illumina’s MiseqPE300 platform.

### 2.6. Statistical Analysis

The data were analyzed by IBM SPSS Statistics 23, using the two-way ANOVA. The interactions of the two factors were tested by the Duncan multiple comparison post hoc test when appropriate. All data were tested for normal distribution and homogeneity before further analysis; the results are expressed as “mean ± standard deviation (SD)”. *p*-values < 0.05 were used to indicate statistical significance.

Alpha and Beta diversity were analyzed with Mothur (v.1.30.2) and displayed with R software (v3.3.1). In addition, the differences in cecal microbiota among groups were compared using the Kruskal–Wallis H test; *p*-values < 0.05 were used to indicate statistical significance. To obtain a more rigorous interpretation, we also use the Bonferroni correction (post hoc test: Tukey–Kramer). Although above the Bonferroni-corrected significance threshold, *p*-values < 0.05 were considered suggestive of evidence for a potential association.

## 3. Results

### 3.1. Effects of LED Light Colors on the Growth Performance

The effect of different color LED lights on the growth performance of the broilers is summarized in Table 1.

The growth performance was significantly affected by age (*p* < 0.001). The BW was significantly higher in the B group than in the G group (*p* < 0.05) at 21 days of age. At day 42 of age, the highest BW was observed in the B and BG groups, while broilers receiving the green light had lowered BW compared with other groups (*p* < 0.05). The ADFI was significantly higher (*p* < 0.05) for 21 d with the blue light compared to the other groups. The difference in ADFI between the B group and W group for 42 d was not significant (*p* > 0.05); the B group was significantly higher than the G and BG groups (*p* < 0.05). At the age of 21 d, the ADG of the B group was significantly higher than the G and BG groups (*p* < 0.05). The ADG was significantly higher in the B and BG groups than in the W and G groups (*p* < 0.05) at 42 d, and the W group was significantly higher than the G group (*p* < 0.05). There was no significant difference (*p* > 0.05) in the FCR across different treatments at 21 d. However, the FCR of the BG group was significantly lower than other groups at 42 d. (*p* < 0.05). There was a significant age × light color effect on BW and ADG (*p* < 0.001).

### 3.2. Effects of LED Light Colors on the Intestinal Morphology

The intestinal histomorphology parameters, such as villus height, crypt depth, and villus/crypt ratio, are shown in Table 2 and Appendix A.

Age causes the villus in the ileum, jejunum, and duodenum to grow higher (*p* < 0.001, *p* < 0.001, *p* = 0.004). The V/C ratio of the duodenum and depth of the jejunum crypt among the four groups were increased at 42 d compared with 21 d (*p* < 0.001, *p* = 0.006).

At 21 d of age, the effects of the four light colors on the morphology of the duodenum, the depth of the jejunal crypt, and the ratio of villus to crypt were insignificant (*p* > 0.05). The villous height of the jejunum and ileum in the B and BG groups were significantly higher than the G group (*p* < 0.05) but did not have a significant difference with the W group (*p* > 0.05), the difference between the W group and G group was not significant (*p* > 0.05). The depth of the ileal recess in the W and B groups was significantly higher than that in the BG group (*p* < 0.05); the BG and G groups did not have a significant difference (*p* > 0.05). The V/C ratio of ileum in the BG group was significantly higher than that in the W and G groups (*p* < 0.05), but there was no significant difference in the B group (*p* > 0.05).

At 42 d of age, a significant difference was found between the B group and W/BG group on the V/C ratio of the duodenum (*p* < 0.05). The ileal recess depth was significantly decreased in the W, G, and BG groups (*p* < 0.05), and the ileal V/C ratio in the G and BG groups was significantly higher than that in the B group (*p* < 0.05). There was no significant difference in the villus height and crypt depth of the duodenum, jejunum, and ileum villus height in broilers with different LED light colors. There was a significant interaction between age and light color treatments on ileal recess depth and V/C ratio (*p*< 0.05).

### 3.3. Effects of LED Light Colors on the Cecal SCFAs Concentrations

The impacts of LED light colors on the cecal SCFA concentrations are presented in Table 3. With increasing broiler age, the cecal short-chain fatty acid concentrations were decreased in the same light environment (*p* < 0.001). On day 21, the concentrations of propionate, isobutyrate, butyrate, and isovalerate in the BG group were significantly higher than those in other groups (*p* < 0.05), but there was no significant difference among G, B, and W groups (*p* > 0.05). The concentration of valerate in the BG group was significantly higher than that in the B and W groups, but there was no significant difference between the BG and G group (*p* > 0.05). On day 42, the concentrations of isobutyrate in the BG group were significantly higher than that in other groups (*p* < 0.05), but there was no significant difference among the G, B, and W groups (*p* > 0.05). There was no significant difference in the concentrations of acetate, propionate, butyrate, isovalerate, and valerate among the four groups. There was a significant interaction between age and light color treatments on propionate, isobutyrate, butyrate, and valerate (*p* < 0.05).

### 3.4. Species Annotation and Assessment in Cecal Microbiota

The 16S rRNA gene amplicon sequences from cecal contents of samples on day 21 and day 42 were conducted to investigate the effects of different light colors on the cecal microbiota of broilers. The results of the day 21 and day 42 analyses show that the rarefaction curve of the Sobs index and Shannon index flattened with increasing sequencing depth, which indicated that our samples covered most microbial species information (Appendix A).

Illumina Miseq sequencing of the 338F_—_806R regions of bacterial 16S rRNA genes generated 10,524,203 and 1,129,642 high-quality sequences at day 21 and day 42, respectively. Four hundred and sixty-five OTUs (day 21) and 618 OTUs (day 42) were obtained, respectively (basis of 97% sequence similarity). As shown in the Venn diagram about day 21, we identified 324, which appeared to be present in all the samples. The W group had 397 OTUs, with 17 unique OTUs; the G group had 395 OTUs, with 11 unique OUTs; the B group had 394 OTUs, with 8 unique OTUs. The BG group has 396 OTUs, with 12 unique OTUs (Figure 2A). On day 42, we observed that 485 OTUs were shared across all treatments. The W group had 579 OTUs, with 5 unique OTUs; the G group had 572 OTUs, with 3 unique OUTs; the B group had 550 OTUs, with 2 unique OTUs. The BG group had 551 OTUs, with 6 unique OTUs (Figure 2B).

The α-diversity mainly reflects the species richness and diversity of the microbial community. Indexes of indexes Shannon, Simpson, Ace, and Chao were calculated as the α-diversity to evaluate the microbial diversity. There were no significant differences in the four indexes among all treatments, whether at 21 d or 42 d. Next, the β-analysis was used to determine the degree of dispersion of the microorganisms between the different light treatment groups; the PCoA analysis was conducted and illustrated based on unweighted Unifrac distances (Figure 3A,B). There were no significant differences in species abundance distribution among the four groups of samples on day 21. On day 42, there were significant differences in β-analysis among the groups.

### 3.5. Effects of LED Light Colors on the Cecal Microbiota at the Phylum Level

A total of nine phyla were detected on the experimental day 21 and eight phyla on day 42, respectively. The dominant phyla bacteria detected (greater than 0.1%) in the cecal contents of 21-day-old broilers mainly included *Bacteroidota* and *Firmicutes*. On day 42, besides *Bacteroidota* and *Firmicutes*, *Cyanobacteria* and *Proteobacteria* were also found to be the dominant bacteria (Figure 4A,B). There was no significant difference in the relative abundance of cecal microorganisms at the phylum level among the four groups on day 21 and day 42.

### 3.6. Effects of LED Light Colors on the Cecal Microbiota at the Genera Level

A total of 118 genera were detected on the experimental day 21 and 133 genera on day 42, respectively. At day 21, 19 genera with relative abundance greater than 0.1% were found (Figure 5A). Except for *Bacteroidetes* (*p* = 0.011, *p*-adjust = 1.000), there was no significant difference in the cecal microbial community at the genus level among the four groups (*p* > 0.05). The relative abundance of *Bacteroides* in the cecum at the genus level of the BG group was significantly higher than that of the G group and the W group (*p* < 0.05, Figure 6A) but not significantly different from the B group (*p* > 0.05, Figure 6A). At day 42, 26 genera with relative abundance greater than 0.1% were found (Figure 5B). Only *Phascolarctobacterium* showed a tendency to differ between the four groups (*p* = 0.006, *p*-adjust = 0.818). The relative abundance of *Phascolarctobacterium* in the cecum at the genus level of the G group was significantly higher than that of the B group and the BG group (*p* < 0.05, Figure 6B), but not significantly different from the W group (*p* > 0.05, Figure 6B). There was no significant difference in *Phascolarctobacterium* among the B group, BG group, and W group (*p* > 0.05, Figure 6B). No *phascolarctobacterium was found in the B group or* the BG group.

### 3.7. Correlations between Cecal Microbiota and SCFAs

The results of day 21 are shown in Figure 7A (for the microorganisms with the total abundance in the top 19). *Bacteroides* is positively correlated with acetate and propionate, while *Lactobacillus* is negatively correlated with acetate and propionate. At the same time, *norank _f_norank_o_Clostridia_UCG-014*, *norank_f_norank_o_RF39*, and *Subdoligranulum* are also negatively correlated with acetate. *Unclassified_ f_Lachnospiraceae* is positively correlated with valerate. The *Ruminoccus_torques_group* is negatively correlated with isobutyrate, butyrate, isovalerate, and valerate.

The results of day 42 were shown in Figure 7B (for the microorganisms with the total abundance in the top 26). *unclassified_f_Lachnospiraceae* and *unclassified_f_Oscillospiraceae* are negatively correlated with acetate. *norank_f_norank_o_Gastranaerophilales* is positively correlated with propionate and valerate. *Blautia* and *Lachnoclostridiumis* are positively correlated with isovalerate, while *Bacteroides* is negatively correlated with isovalerate. *Blautia* is positively correlated with butyrate.

## 4. Discussion

### 4.1. Growth Performance

Light color plays a crucial role in affecting the growth performance of broilers. The eyes of birds can sense a wider wavelength range compared to mammals. Therefore, poultry is more sensitive to light stimuli. Many studies revealed that colored light promotes broiler growth more than white light, and short wavelength (blue light and green light) stimulates growth rapidly [12,19]; long wavelength (red light and orange light) accelerates development and sexual maturation [20]. In the current study, the B group had the highest ADFI at both 21 d and 42 d, indicating that blue light could promote broiler feed intake, thereby improving their growth and development.

At the age of 21 days, the feed intake of the BG group was significantly lower than that of the B group, but the final weight was maintained at a similar level to that of the B group, indicating that the blue–green composite light may have better nutrient metabolism levels for 21-day-old broilers. At the age of 42 days, the F/G of the BG group was significantly lower than that of other experimental groups, and the advantage of the BG group on the production performance of 42-day-old broilers was better than that of the W and G groups, which represents the advantage of blue–green composite light to reduce F/G and improve broiler performance. It has been shown that combining the advantages of monochromatic light can affect the growth of broilers [11,21]. Similarly, the blue-green composite light selected in this experiment realized the growth promotion of broilers.

### 4.2. Intestinal Morphology

The small intestine plays a significant role in digestion and absorption, and its development directly affects the growth of the body. Villus height, crypt depth, and V/C ratio can reflect intestinal health and functional status [22,23]. Therefore, a high V/C ratio can also indicate that the intestinal digestion and absorption capacity is strong. Early-life gut well can maximize the impact on the growth potential of broilers [24]. Thus, early-life gut morphology and function are beneficial for broiler growth performance [25]. The results of this study indicate that the height of small intestinal villi increased with age in the same light color environment. Liao et al. suggested that the villous height in the duodenum, jejunum, and ileum all increased with age [26]. The results were similar to ours.

Previous studies have suggested that blue light and green light enhanced the growth of villus height [27], as Yang et al. [28] reported that blue light enhanced the digestive and absorptive capacity of the jejunum and increased goblet cells, villus height, and V/C ratios. Our study showed an increase in the villus height of the jejunum at 21 d and V/C ratios of the duodenum at 42 d in the B group. Intestinal morphology is an indicator of intestinal health, and its values indicate digestive and absorptive capacity. The proximal small intestine is the main site for the digestion and absorption of nutrients like fat, protein, and sugars [29]. Therefore, our results suggest that nutrient uptake enhancement in the small intestine and broiler weight gain in the B group matched the results obtained by growth performance.

Meanwhile, the results of this study demonstrated that the blue–green composite light significantly reduced crypt depth and increased V/C ratios of ileal in broilers, which the blue–green composite light has the effect of improving the intestinal morphology of the ileum of broilers either 21 d or 42 d. The higher the V/C value, the stronger the digestion and absorption capacity. Although the feed intake of the BG group was significantly lower than that of the B group, the FBW could be maintained at a level similar to that of the B group, which may be related to good intestinal development. A well-developed gut can facilitate nutrient absorption. In addition, studies show that mixing blue and green light can combine the advantages of both wavelengths, showing better stimulation than monochromatic light [28]. Our results are in agreement with previous studies that the blue-green composite light is beneficial for broilers. In brief, it can promote better intestine in broilers.

### 4.3. Cecal SCFAs Concentrations

SCFAs are metabolites produced by intestinal microbiota. It plays an important role in the energy metabolism of the host, which can provide nutrition for intestinal cells, thus maintaining epithelial barrier function, regulating epithelial hyperplasia, maintaining intestinal immune homeostasis, and promoting skeletal muscle growth, thus maintaining the health of livestock and poultry in an all-round way and promoting their growth and development [30,31,32]. Experiments on mice and humans have shown that SCFAs decline with age [33,34], and our results also present significantly lower SCFAs at 42 d than at 21 d in each group. This is in contrast to the findings of Liao et al. [26], who argued that most cecal SCFAs increased, but isobutyrate decreased with age.

In this study, the results showed that blue–green composite light could significantly promote the production of propionate, isobutyrate, butyrate, isovaleric, and valerate in the cecum of 21-day-old broilers. Based on the growth performance data of 21-day-old broilers, the results showed that the high concentrations of SCFAs in the cecum of broilers in the BG group promoted higher energy acquisition, which was beneficial to the intestinal development, maintained intestinal health, and promoted body growth. Isobutyrate also belongs to branched-chain fatty acids (BCFAs) [35]. There are differences in the identification of BCFAs. According to some studies, the BCFAs can lead to high protein catabolism, which leads to a high obesity rate. The increase of BCFAs in the colon indicates that it is not conducive to nutritional metabolism [36]. However, in rat models, Yan et al. found that BCFAs could reduce the incidence of necrotizing enterocolitis, increase the content of IL-10 in intestinal anti-inflammatory cells, and change the microbial ecology of the gastrointestinal tract [37]. The blue–green composite light could promote the production of isobutyrate in the cecum of broilers so as to protect the intestinal barrier, improve the absorption of nutrients, and promote growth. This result was consistent with the growth performance data of 42-day-old broilers; the broilers in the BG group had less food intake but higher weight gain. And the results of the interaction proved that the effect of age on propionate, isobutyrate, butyrate, and valerate were influenced by light color. It is possible that this is the cause of the difference between our results and those of Liao et al. [26].

### 4.4. Cecal Microbiota

The gut microbiota is an important factor in promoting animal growth and development and changes with the influence of the surrounding environment, diet, circadian rhythm, and other factors, but the core microbiota of adult animals remains stable [38,39]. For broilers, the differences in the microbiome were no longer significant after 14 days of age, and the gut microbiome reached a plateau at 21 days of age [40,41]. Most of the current research results show that *Firmicutes* and *Bacteroidota* are the two phyla with the highest relative abundance in the gut of broilers [42]. Consistent with the results of most studies, in the analysis of 21-day-old broiler caecum microbial composition, it was found that *Bacteroidota* and *Firmicutes* were the most abundant in cecal content samples of broiler chickens raised under different LED light colors and remained the most abundant until 42 days of age. *Bacteroidota* and *Firmicutes* were also relatively stable in abundance among groups. In experiments with mice, aging was accompanied by an increase in the abundance of Proteobacteria and cyanobacteria in the gut [43]. Compared with day 21, this experiment also found that *Proteobacteria* and *Cyanobacteria* were increased in the caeca of broilers at day 42.

*Bacteroidetes* can produce SCFAs by breaking down carbohydrates, and these SCFAs are absorbed by the cecal epithelium into the blood circulation to provide energy for the body [40]. The SCFAs’ and Bacteroidetes’ relative abundance of the cecum in the BG group was higher at the age of 21 days, and through correlation analysis, we found that Bacteroides are positively correlated with acetate and propionate. The results indicate that the cecal microorganisms of the broiler groups may affect the SCFAs’ production. Zhang et al. found that perfusion of sodium propionate could inhibit the feed intake of growing pigs but can increase the ratio of villous crypt in the jejunum [44]. The results of this experiment showed a similar phenomenon. Therefore, the lower feed intake of broilers in the BG group might be due to the high cecal propionate. But, the BG group had better intestinal development and finally reached a level similar weight to that in the B group. On day 42, *Phascolarctobacterium* was only found in the white light group and the green light group. Studies have shown that body weight and fat mass negatively correlate with Phascolarctobacterium abundance [45,46]. The W group and the G group had lower body weight, which may be related to the presence of *Phascolarctobacterium*. The results of the correlation analysis showed that there was a correlation between cecal microorganisms and SCFAs. However, there was no significant correlation between the differential SCFAs (Isobutyrate) and the differential cecal microorganisms (*Phascolarctobacterium*) in the four groups, which might be due to the existence of unknown intermediate cecal microorganisms and their metabolites in the regulatory process, the specific mechanism needs further systematic and in-depth experimental investigation.

*Bacteroidetes* are from the phylum *Bacteroidota*, and *Phascolarctobacterium* is from the phylum *Firmicutes*. The elevated ratio of *Firmicutes* and *Bacteroides* is associated with fat deposition and affects SCFA production and carbohydrate metabolism [47]. Compared with the W group and the G group, the B group and the BG group had a higher abundance of *Bacteroides* at 21 days of age and lower Firmicutes at 42 days of age, so the ratio of *Firmicutes* and *Bacteroides* decreased in the BG group and B group, which was beneficial to improve fat deposition and promote the production of SCFAs. Therefore, the blue light and the blue–green composite light may improve the growth performance by affecting the bacterial groups involved in the SCFA metabolic pathway.

It is worth noting that there is a possibility of a false negative for the Bonferroni-corrected test. Our study revealed that *Bacteroidetes* and *Phascolarctobacterium* are affected by light colors. However, these significances disappeared after the Bonferroni-corrected test. We suppose that this is an exploratory study; the unadjusted *p*-values suggest potential significance levels, which should be validated in future studies.

## 5. Conclusions

Our results demonstrated that the effects of light colors on growth performance, intestinal morphology, cecal SCFAs, and microbiota in the broilers were carried out in this study. We observed that broilers reared in the blue light could promote broiler feed intake, thereby improving their growth and development, and the blue–green composite light has the advantage of reducing F/G and improving broiler performance. Therefore, our results suggest that nutrient uptake enhancement in the small intestine and broiler weight gain with blue light and blue–green composite light can improve the morphology of the ileum in broilers, promoting better intestinal health of broilers.

The further analysis of the cecal microbiota among four groups at two experimental points (day 21 and day 42) revealed that blue light and blue–green composite light can affect SCFA concentrations by changing the abundance of genus-level flora and improving growth performance. Of course, this requires further experiments to validate and explore the mechanisms involved.

## Figures and Tables

**Figure 1 animals-13-03731-f001:**
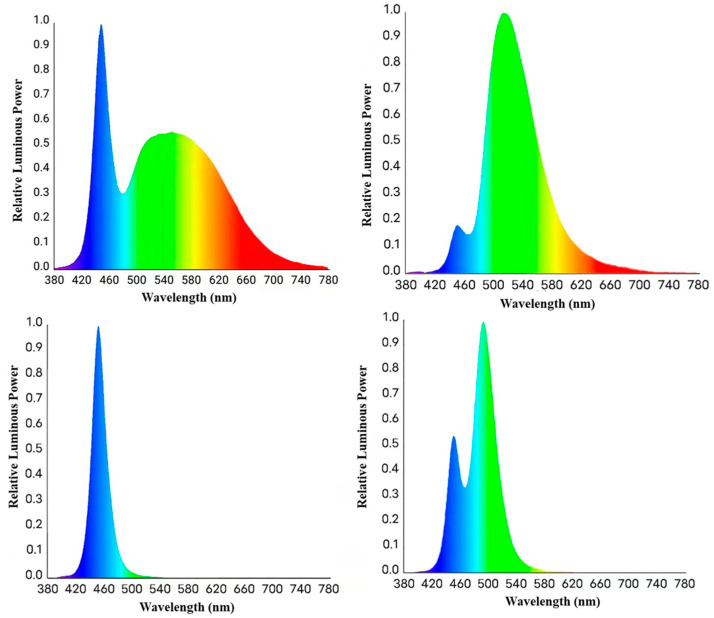
Spectral characteristics: (**A**) white light-emitting diode light (WL); (**B**) green light-emitting diode light (GL); (**C**) blue light-emitting diode light (BL); (**D**) blue-green light-emitting diode light (BGL).

**Figure 2 animals-13-03731-f002:**
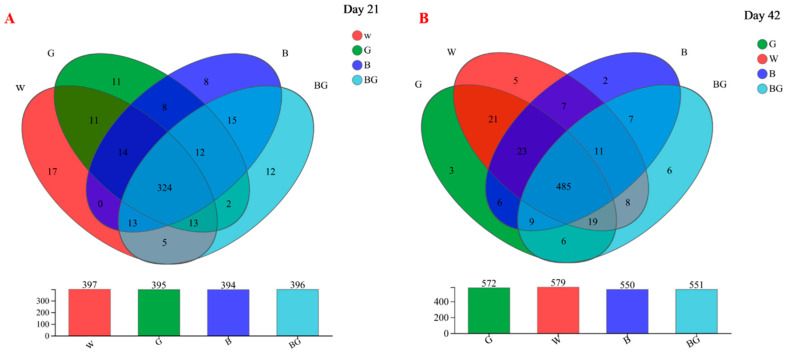
Venn diagrams of cecal microbiota at day 21 (**A**) and at day 42 (**B**).

**Figure 3 animals-13-03731-f003:**
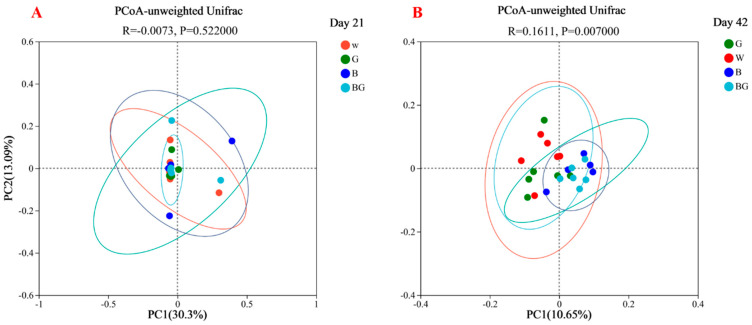
Effects of LED light colors on the β-analysis of microbiota at day 21 (**A**) and at day 42 (**B**). *p >* 0.05 indicates no significant difference, and *p* < 0.05 indicates a significant difference.

**Figure 4 animals-13-03731-f004:**
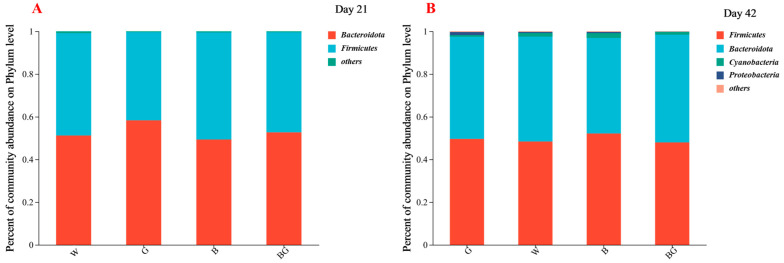
Effects of LED light colors on the cecal microbiota compositions at the phylum level on day 21 (**A**) and on day 42 (**B**).

**Figure 5 animals-13-03731-f005:**
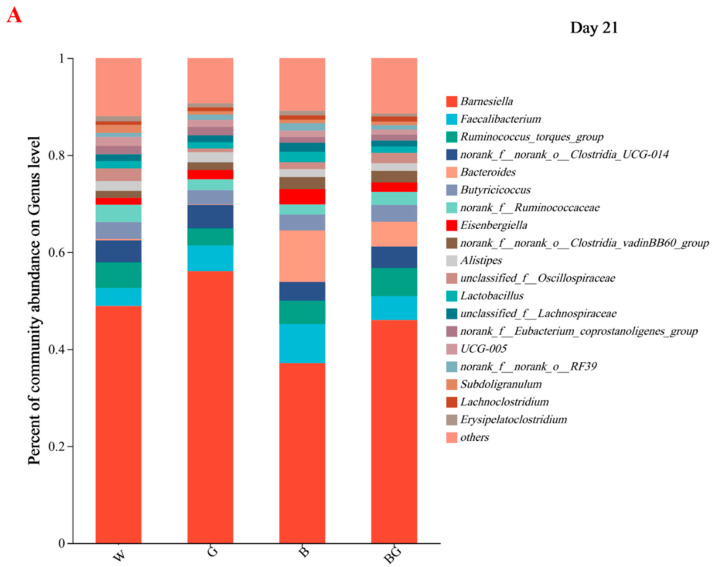
Effects of LED light colors on the cecal microbiota compositions at the genus level on day 21 (**A**) and on day 42 (**B**).

**Figure 6 animals-13-03731-f006:**
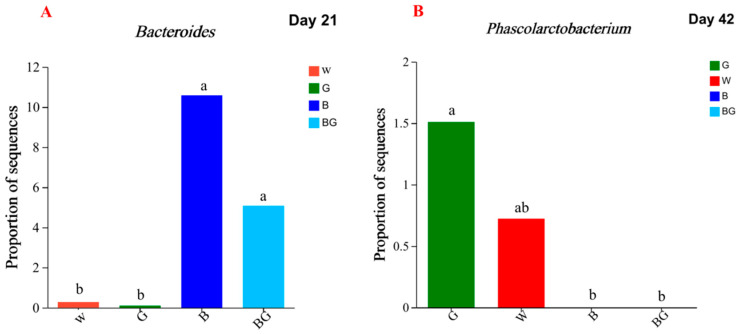
Effects of LED light colors on the cecal microbiota abundance at the genus level on day 21 (**A**) and on day 42 (**B**). ^a,b^ Significant effect of treatment (*p* < 0.05; values with different lowercase letters are significantly different).

**Figure 7 animals-13-03731-f007:**
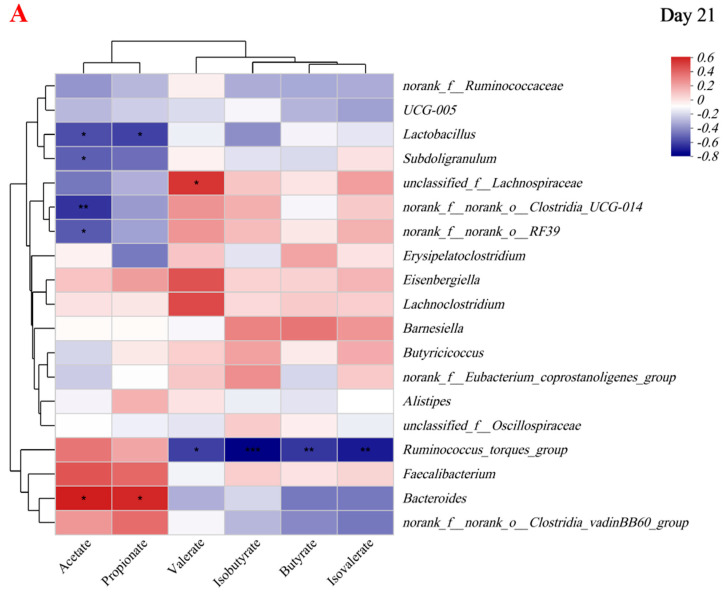
Heatmap of the Spearman correlations between the cecal microbiota and short-chain fatty acid on day 21 (**A**) and on day 42 (**B**). * *p* < 0.05 (Significant); ** *p* < 0.01 (Extremely significant); *** *p* < 0.001 (The most significant).

**Table 1 animals-13-03731-t001:** Effects of broilers’ growth performance with the different LED lights.

Items	White Light	Green Light	Blue Light	Blue-Green Light	*p*-Value		
					Age	Light Color	Age × Light Color
BW ^1^/g							
21 d	806.03 ± 27.57 ^Bab^	779.92 ± 23.34 ^Bb^	822.97 ± 5.59 ^Ba^	796.29 ± 16.80 ^Bab^	<0.001	<0.001	<0.001
42 d	2338.89 ± 75.78 ^Ab^	2212.68 ± 29.61 ^Ac^	2492.53 ± 74.23 ^Aa^	2456.06 ± 119.67 ^Aa^			
ADFI ^2^/g							
21 d	48.73 ± 1.30 ^Bb^	48.13 ± 1.08 ^Bb^	51.06 ± 0.68 ^Ba^	48.19 ± 2.25 ^Bb^	<0.001	<0.001	0.111
42 d	82.72 ± 2.16 ^Aab^	77.45 ± 4.94 ^Ab^	86.28 ± 4.11 ^Aa^	78.83 ± 5.20 ^Ab^			
ADG ^3^/g							
21 d	36.20 ± 1.22 ^Bab^	34.95 ± 1.56 ^Bc^	37.19 ± 0.48 ^Ba^	35.48 ± 0.43 ^Bbc^	<0.001	<0.001	<0.001
42 d	54.59 ± 1.78 ^Ab^	51.59 ± 0.69 ^Ac^	58.25 ± 1.78 ^Aa^	57.38 ± 2.81 ^Aa^			
FCR ^4^							
21 d	1.37 ± 0.21 ^B^	1.34 ± 0.05 ^B^	1.37 ± 0.12 ^B^	1.35 ± 0.06 ^B^	<0.001	0.028	0.079
42 d	1.52 ± 0.06 ^Aa^	1.50 ± 0.09 ^Aa^	1.48 ± 0.10 ^Aa^	1.38 ± 0.09 ^Ab^			

Mean values with different superscripts (a–c) in the same row differ within light color treatments (*p* < 0.05). Mean values with different superscripts (A,B) in the same column differ within Age (*p* < 0.05). ^1^ BW: The final body weight; ^2^ ADFI: Average daily feed intake; ^3^ ADG: Average daily gain; ^4^ FCR: Feed conversion rate.

**Table 2 animals-13-03731-t002:** Effects of different LED light colors on intestinal morphology of broilers.

Items	Stage	White Light	Green Light	Blue Light	Blue–Green Light	*p*-Value		
						Age	Light Color	Age × Light Color
Duodenum								
Villus Height/μm	21 d	1016.82 ± 163.83 ^B^	957.12 ± 140.73 ^B^	1023.87 ± 133.83 ^B^	1056.44 ± 26.92 ^B^	<0.001	0.559	0.949
	42 d	1300.90 ± 219.90 ^A^	1258.40 ± 71.89 ^A^	1380.93 ± 219.18 ^A^	1343.38 ± 216.58 ^A^			
Crypt Depth/μm	21 d	96.77 ± 4.62	83.14 ± 6.45	94.59 ± 16.54	102.06 ± 22.79	0.428	0.063	0.521
	42 d	99.36 ± 17.77	86.12 ± 10.23	85.76 ± 10.32	92.72 ± 5.61			
V/C ^1^	21 d	10.51 ± 1.19 ^B^	12.62 ± 3.00 ^B^	11.46 ± 2.54 ^B^	10.78 ± 2.74 ^B^	<0.001	0.008	0.146
	42 d	12.14 ± 1.00 ^Ac^	15.45 ± 1.17 ^Aab^	17.40 ± 2.70 ^Aa^	14.18 ± 0.45 ^Abc^			
Jejunum								
Villus Height/μm	21 d	957.39 ± 95.65 ^Bab^	809.95 ± 91.03 ^Bb^	1103.88 ± 174.57 ^Ba^	992.64 ± 168.35 ^Ba^	<0.001	0.038	0.363
	42 d	1301.59 ± 279.39 ^A^	1076.72 ± 230.86 ^A^	1178.98 ± 57.39 ^A^	1244.10 ± 117.57 ^A^			
Crypt Depth/μm	21 d	75.76 ± 5.94 ^B^	76.13 ± 7.49 ^B^	83.84 ± 9.59 ^B^	75.38 ± 7.06 ^B^	0.006	0.037	0.989
	42 d	81.71 ± 2.13 ^A^	83.30 ± 5.99 ^A^	89.00 ± 7.78 ^A^	81.70 ± 6.12 ^A^			
V/C	21 d	12.74 ± 1.93	11.61 ± 2.33	13.39 ± 2.99	13.36 ± 3.12	0.193	0.266	0.416
	42 d	16.12 ± 3.88	12.61 ± 3.53	12.64 ± 2.71	14.28 ± 2.90			
Ileum								
Villus Height/μm	21 d	805.18 ± 64.25 ^Bab^	744.07 ± 40.46 ^Bb^	836.21 ± 75.90 ^Ba^	873.58 ± 40.13 ^Ba^	0.004	0.184	0.255
	42 d	879.85 ± 224.98 ^A^	971.87 ± 134.63 ^A^	857.75 ± 62.91 ^A^	1017.64 ± 150.49 ^A^			
Crypt Depth/μm	21 d	100.64 ± 16.21 ^a^	80.05 ± 11.41 ^bc^	87.90 ± 9.39 ^ab^	72.09 ± 3.07 ^c^	0.614	0.001	0.002
	42 d	82.08 ± 6.00 ^b^	83.18 ± 11.27 ^b^	98.32 ± 3.80 ^a^	82.98 ± 5.08 ^b^			
V/C	21 d	8.55 ± 1.47 ^b^	9.43 ± 1.20 ^b^	10.04 ± 2.04 ^ab^	11.83 ± 0.67 ^a^	0.076	0.002	0.044
	42 d	10.72 ± 2.66 ^ab^	11.73 ± 1.34 ^a^	8.69 ± 0.55 ^b^	12.31 ± 2.00 ^a^			

Mean values with different superscripts (a–c) in the same row differ within light color treatments (*p* < 0.05). Mean values with different superscripts (A,B) in the same column differ within Age (*p* < 0.05). There is no significant difference in unmarked letters (*p* > 0.05). ^1^ V/C: Villus height/Crypt depth ratio.

**Table 3 animals-13-03731-t003:** Effects of different LED light colors on cecal short-chain fatty acid concentrations of broilers.

Items	White Light	Green Light	Blue Light	Blue–Green Light	*p*-Value		
					Age	Light Color	Age × Light Color
Acetate, mmol/L							
21 d	61.85 ± 8.51 ^A^	66.64 ± 5.09 ^A^	64.79 ± 13.67 ^A^	79.93 ± 25.72 ^A^	<0.001	0.174	0.308
42 d	16.24 ± 6.20 ^B^	26.59 ± 10.64 ^B^	17.69 ± 0.42 ^B^	19.99 ± 9.02 ^B^			
Propionate, mmol/L							
21 d	15.23 ± 1.04 ^Ab^	17.49 ± 1.26 ^Ab^	17.21 ± 5.31 ^Ab^	26.66 ± 9.13 ^Aa^	<0.001	0.009	0.002
42 d	3.31 ± 1.85 ^B^	3.77 ± 2.18 ^B^	2.48 ± 0.35 ^B^	2.59 ± 1.43 ^B^			
Isobutyrate, mmol/L							
21 d	2.93 ± 0.25 A^b^	3.35 ± 0.12 ^Ab^	3.04 ± 0.24 ^Ab^	5.05 ± 1.52 ^Aa^	<0.001	<0.001	0.004
42 d	0.37 ± 0.18 ^Bb^	0.53 ± 0.21 ^Bb^	0.38 ± 0.05 ^Bb^	0.88 ± 0.24 ^Ba^			
Butyrate, mmol/L							
21 d	6.60 ± 0.59 ^Ab^	7.48 ± 0.85 ^Ab^	6.61 ± 1.10 ^Ab^	10.09 ± 3.04 ^Aa^	<0.001	0.006	0042
42 d	1.85 ± 1.02 ^B^	2.14 ± 1.02 ^B^	2.21 ± 0.40 ^B^	2.44 ± 1.12 ^B^			
Isovalerate, mmol/L							
21 d	2.55 ± 0.25 ^Ab^	3.01 ± 0.16 ^Ab^	2.82 ± 0.49 ^Ab^	4.12 ± 1.22 ^Aa^	<0.001	<0.001	0.051
42 d	0.35 ± 0.16 ^B^	0.37 ± 0.18 ^B^	0.43 ± 0.07 ^B^	0.71 ± 0.46 ^B^			
Valerate, mmol/L							
21 d	2.36 ± 0.19 ^Abc^	2.57 ± 0.14 ^Aab^	2.29 ± 0.18 ^Ac^	2.70 ± 0.06 ^Aa^	<0.001	0.077	0.014
42 d	0.33 ± 0.10 ^B^	0.27 ± 0.15 ^B^	0.30 ± 0.02 ^B^	0.28 ± 0.13 ^B^			

Mean values with different superscripts (a–c) in the same row differ within light color treatments (*p* < 0.05). Mean values with different superscripts (A,B) in the same column differ within Age (*p* < 0.05).

## Data Availability

The data presented in this study are available on request from the first author/corresponding authors.

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
