# Peer review of "Effects of LED Light Colors on the Growth Performance, Intestinal Morphology, Cecal Short-Chain Fatty Acid Concentrations and Microbiota in Broilers"

_animals, 2023, doi:10.3390/ani13233731_

Round 1
Reviewer 1 Report
Comments and Suggestions for Authors
Manuscript entitled ( Effects of LED Light Colors on the Growth Performance, Intestinal Morphology, Cecal Short-chain Fatty Acid Concentrations and Microbiota in Broilers) can't be accepted because it's old and many many reports were conducted concerning this subject, no novelty presented
Author Response
Dear reviewer:
Thank you very much for the review and comments on our manuscript. We have responded to each of your comments and suggestions in the document, so please review it.

Reviewer 2 Report
Comments and Suggestions for Authors
Dear Author,
This study investigated the effects of different colors of light on broiler performance, intestinal morphology, cecal fatty acid concentration, and microbial content. There are previous studies on similar subjects (https://doi.org/10.2141/jpsa.0130049). The current study assessed the fatty acid composition and microbial content differently than the previous study. Therefore, it can be stated that the study is original.
Introduction
It should be stated in which species the studies were performed.
The deficiencies that the current study will try to solve with regard to this topic should be stated in a paragraph.
Materials and Methods
Stocking density should be specified.
In the statistics section, should the values in the tables be stated as standard error or standard deviation?
Results
Table 4. Delete lines 83-84: “a - c Significant effect of treatment ………”, “This sentence was written even though the difference between the groups was not significant.”
Lines 156-157: This sentence should be rewritten. Some between-group differences appear statistically insignificant.
Figure 3: Delete Lines 65-66. “There is no significant difference in unmarked letters (P > 0.05), a - c Significant effect of treatment ……” ,“This sentence was written even though the difference between the groups was not significant.”
Figure 7: Lettering should be done in the figure. While there was a significant difference between the groups, no lettering was performed.
The lettering should be applied properly on the Figure 7.
The conclusion part is sufficient.
References are sufficient and appropriate.
Author Response

(The authors gave the same response as above.)

Reviewer 3 Report
Comments and Suggestions for Authors
The paper test the effect of colored light on broilrs growth, intestine development and micribiota.
Major problem in M&M, were written that birds were killed by cervical dislocation. According to FASS guid line for the use of farm animals in research you are allowed to do cervical dislocation up to 2kg BW. In this experiment birds were killed at 42 d of age and BW was above 2 kg.
2.1 Animals: Are animals grow according to primary breeder recommendations?
Fig 1: Green light is not pure monochromatic, sorry it is contaminated lamp that might affect the results.
Brightness of 20 lux is problematic especially in the blue light a careful examination should be done by measuring intensity in watts/m2
2.2 fasting broilers is huge stress, it is not a common practice in broiler farming. Either explain why? Or provide ref from primary breeders to this kind of protocol.
2.3 please write avg Bw + SE . it is nor correct to write “whose body weight was close to avg..”
2.6 Statistics should run 2 way analysis Light X age
3.1 BW at 42 days are low (fasting effect)
Other results show a lot of non-significant results (table 2 can be removed, just mention the important findings, Fig 3 and Table 4 remove)
Author Response

(The authors gave the same response as above.)

Round 2
Reviewer 1 Report
Comments and Suggestions for Authors
Thanks for authors
Author Response
Dear reviewer:
Thank you very much for the review and comments on our manuscript.
Reviewer 3 Report
Comments and Suggestions for Authors
I have already suggested
Author Response
Dear Reviewer,
Thank you very much for the review and comments on our manuscript. The comments are all valuable for revising and improving our paper. We performed a two-factor analysis as you suggested and did our best to address the other issues you raised. And the P-values in the microbiological analysis section were Bonferroni corrected according to academic editor suggestion. All the changes made in the text were marked in Yellow so that they can be quickly identified. We believe that the revised manuscript is much better than the former one. We hope that the revised manuscript is now suitable for publication in ANIMALS. Thank you!